# CMCL 2022 Shared Task on Multilingual and Crosslingual Prediction of Human Reading Behavior

**Nora Hollenstein**
University of Copenhagen
nora.hollenstein@hum.ku.dk

**Emmanuele Chersoni**
The Hong Kong Polytechnic University
emmanuelechersoni@gmail.com

**Cassandra Jacobs**
University of Buffalo
jacobs.cassandra.l@gmail.com

**Yohei Oseki**
University of Tokyo
oseki@g.ecc.u-tokyo.ac.jp

**Laurent Prévot**
Aix-Marseille Université & CNRS, LPL
laurent.prevot@univ-amu.fr

**Enrico Santus**
Bayer Pharmaceuticals
esantus@gmail.com

## Abstract

We present the second shared task on eye-tracking data prediction of the Cognitive Modeling and Computational Linguistics Workshop (CMCL). Differently from the previous edition, participating teams are asked to predict eye-tracking features from multiple languages, including a surprise language for which there were no available training data. Moreover, the task also included the prediction of standard deviations of feature values in order to account for individual differences between readers.

A total of six teams registered to the task. For the first subtask on multilingual prediction, the winning team proposed a regression model based on lexical features, while for the second subtask on cross-lingual prediction, the winning team used a hybrid model based on a multilingual transformer embeddings as well as statistical features.

## 1 Introduction

The benefits of eye movement data for machine learning have been assessed in various domains, including NLP (Barrett et al., 2016, 2018; McGuire and Tomuro, 2021) and computer vision (Shanmuga Vadivel et al., 2015; Kruthiventi et al., 2017; Bautista and Naval, 2020; Tseng et al., 2020). Eye-tracking provides millisecond-accurate records on where humans look when they are reading and are useful in explanatory research of language processing. Eye movements depend on the stimulus and are therefore language-specific, but there are also universal tendencies that have been observed across languages (Liversedge et al., 2016).

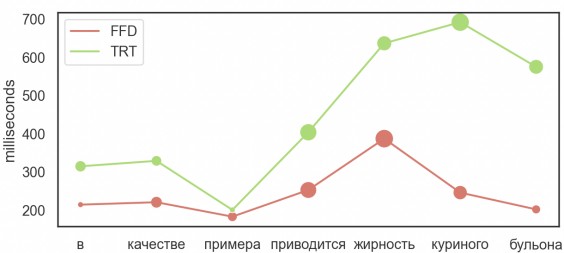

Figure 1: An example sentence from the Russian Sentence Corpus (Laurinavichyute et al., 2019) averaged across all readers. A wider diameter of the markers represents a higher standard deviation.

Modelling human reading has been researched extensively in psycholinguistics (Reichle et al., 1998; Matthies and Søgaard, 2013; Hahn and Keller, 2016). In NLP, eye-tracking prediction has been used to determine linguistic complexity (Singh et al., 2016; Sarti et al., 2021) or to analyze language models' ability to account for measures of human reading effort (Merkx and Frank). Being able to accurately predict eye-tracking features across languages will advance this field and will facilitate comparisons between models and the analysis of their varying capabilities.

In this shared task, we address the challenge of predicting eye-tracking features recorded during sentence processing of multiple languages. We are interested in both cognitive modelling approaches as well as linguistically motivated approaches (i.e., language models). This shared task is hosted on CodaLab, where the instructions and pre-processed

eye-tracking datasets are available.[1]

Compared to the CMCL 2021 Shared Task on eye-tracking prediction (Hollenstein et al., 2021a), we introduce two major changes:

- Multilingual data: We provide an eye movement dataset with sentences from six different languages (Chinese, Dutch, English, German, Hindi, Russian) for Subtask 1 and a new Danish test set for Subtask 2.

- Eye-tracking features: To take into account the individual differences between readers, the task is not limited to predict the mean eye tracking features across readers, but also the standard deviation of the feature values.

## 2 Related Work

### 2.1 Eye-Tracking and Language Models

It is widely acknowledged by researchers on naturalistic reading that fixation patterns are influenced by the words' contextual predictability (Ehrlich and Rayner, 1981), although there is some substantial disagreement about the nature of this link (Brothers and Kuperberg, 2021). In Natural Language Processing, the most influential account of this phenomenon comes from *surprisal theory* (Hale, 2001; Levy, 2008). This theory claims that the processing difficulty of a word is proportional to its *surprisal*, i.e., the negative logarithm of the probability of the word given the context, and it served as a reference framework for several studies on language models and eye-tracking data prediction (Demberg and Keller, 2008; Frank and Bod, 2011; Fossum and Levy, 2012). Surprisal is not necessarily the only factor involved: for example, word length, word frequency, and other local statistics (e.g., bigram and trigram probabilities) also affect reading times (Rayner and Raney, 1996; Williams and Morris, 2004; Goodkind and Bicknell, 2021). Embedding-based semantic similarity was also found to be correlated with eye-tracking metrics (Mitchell et al., 2010; Salicchi et al., 2021; Yu et al., 2021), although it is not clear whether its effect is independent of surprisal (Frank, 2017).

Later research work brought evidence that language models with a lower perplexity are better at fitting to human reading times (Goodkind and Bicknell, 2018; Aurnhammer and Frank, 2019; Wilcox et al., 2020; Merkx and Frank). However, other studies suggested that perplexity may not tell the whole story. For example, Hao et al. (2020) pointed out that such a metric cannot be used for comparing models with different vocabularies and proposed, as a more reliable predictor, the correlation between surprisal values computed by a language model and the surprisal values obtained from humans by means of a Cloze test. Moreover, while most work on eye-tracking and language modeling focused on English, recent experiments on typologically distant languages like Japanese showed that lower-perplexity models may not be necessarily better at predicting eye-movement data (Kuribayashi et al., 2021). Therefore, multilingual evaluation is an important step for building cognitively plausible models of human reading processes.

### 2.2 Multilingual Eye-Tracking Corpora

Comparing monolingual and multilingual Transformer models, Hollenstein et al. (2021b) found that the latter are surprisingly accurate in predicting eye-tracking features across languages. In particular, multilingual BERT (Devlin et al., 2019) shows the best crosslinguistic transfer ability, even without being explicitly trained on the target language, while the XLM models (Lample and Conneau, 2019) achieve better in-language performance after fine-tuning.

Psycholinguistic research in the last two decades has led to the introduction of corpora with eye-tracking recordings in several languages, including English (Cop et al., 2017; Luke and Christianson, 2017; Hollenstein et al., 2018, 2020), German (Kliegl et al., 2006; Jäger et al., 2021), Hindi (Husain et al., 2015), Japanese (Asahara et al., 2016), Dutch (Cop et al., 2017), Russian (Laurinavichyute et al., 2019), Mandarin Chinese (Pan et al., 2021), and Danish (Hollenstein et al., 2022). However, it is not optimal to combine datasets recorded in different settings. The most recent release is the Multilingual Eye-Movement Corpus (MECO; Siegelman et al. 2022), a new resource including parallel data from 580 readers of 13 different languages following the same experiment protocol. The notable differences between these corpora and other psycholinguistic studies is the naturally occurring stimuli, the presentation of full sentences or longer text spans, and that the participants were able to read in their own speed.

[1]https://competitions.codalab.org/competitions/36415

| Corpus | Lang. | Sents. | Tokens | Subjects | Reference |
|---|---|---|---|---|---|
| BSC | ZH | 150 | 1685 | 60 | Pan et al. (2021) |
| PAHEC | HI | 153 | 2596 | 30 | Husain et al. (2015) |
| RSC | RU | 144 | 1417 | 102 | Laurinavichyute et al. (2019) |
| Provo | EN | 189 | 2659 | 84 | Luke and Christianson (2017) |
| ZuCo 1.0 | EN | 300 | 6588 | 12 | Hollenstein et al. (2018) |
| ZuCo 2.0 | EN | 349 | 6828 | 18 | Hollenstein et al. (2020) |
| GECO-NL | NL | 800 | 9218 | 18 | Cop et al. (2017) |
| PoTeC | DE | 101 | 1895 | 75 | Jäger et al. (2021) |
| CopCo (*Subtask 2 only*) | DK | 402 | 6768 | 5 | Hollenstein et al. (2022) |

Table 1: Datasets used in the shared task. Note that the number of sentences and tokens refers to the text materials we have selected and not necessarily to the complete original datasets.

| Feature | min | max | mean (std) |
|---|---|---|---|
| FFDAVG | 0.0 | 56.74 | 13.02 (7.34) |
| FFDSTD | 0.0 | 58.54 | 4.47 (3.55) |
| TRTAVG | 0.0 | 100.0 | 18.87 (11.57) |
| TRTSTD | 0.0 | 100.0 | 9.86 (8.01) |

Table 2: Minimum, maximum, mean and standard deviation of the *scaled* feature values in both training and test data of Subtask 1, after averaging across readers.

## 3 Task Description

The shared task is formulated as a regression task to predict 2 eye-tracking features and the corresponding standard deviation across readers for each word:

1. FFDAVG: first fixation duration (FFD), the duration of the first fixation on the prevailing word;

2. FFDSTD: standard deviation of FFD across readers;

3. TRTAVG: total reading time (TRT), the sum of all fixation durations on the current word, including regressions;

4. TRTSTD: standard deviation of TRT across readers.

### 3.1 Subtask 1

The goal of the first subtask is multilingual eye tracking prediction, i.e., to predict the eye-tracking features for sentences of the 6 provided languages in the training data on held-out sentences of the same languages in the test data. The dataset contains sentences from a range of openly available eye-tracking corpora.

### 3.2 Subtask 2

The second subtask test the models' performances in a cross-lingual prediction scenario. The training and development data are identical to Subtask 1, but the test data contains eye-tracking data from a new language. The participants were only informed about which language would be included in this subtask at the beginning of the evaluation phase.

## 4 Data

### 4.1 Subtask 1

The dataset contains sentences from the following openly available eye-tracking corpora: the Beijing Sentence Corpus (BSC; Pan et al. 2021), the Postdam-Allahabad Hindi Eye Tracking Corpus (PAHEC; Husain et al. 2015), the Russian Sentence Corpus (RSC; Laurinavichyute et al. 2019), the Provo Corpus (Luke and Christianson, 2017), the Zurich Cognitive Language Processing Corpus (ZuCo; Hollenstein et al. 2018, 2020), The Dutch part of the Ghent Eye-Tracking Corpus (GECO-NL; Cop et al. 2017), and the Potsdam Textbook Corpus (PoTeC; Jäger et al. 2021). These datasets cover a diverse range of text domains, including news articles, novels, Wikipedia sentences, scientific textbook passages, etc. The details are presented in Table 1.

The training data contains 1703 sentences, the development set contains 104 sentences, and the test set 324 sentences.

### 4.2 Subtask 2

As described, the training and development data are identical to Subtask 1, but the test data contains eye-tracking data from a new language, namely Danish. The Danish eye-tracking data contains 402

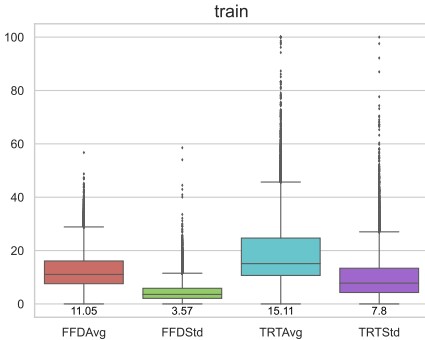

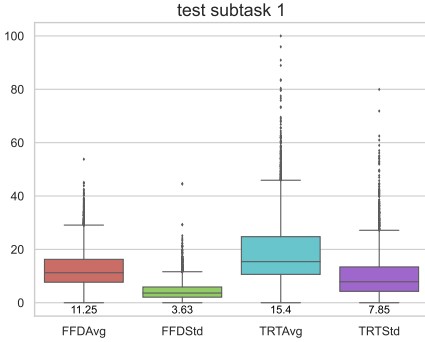

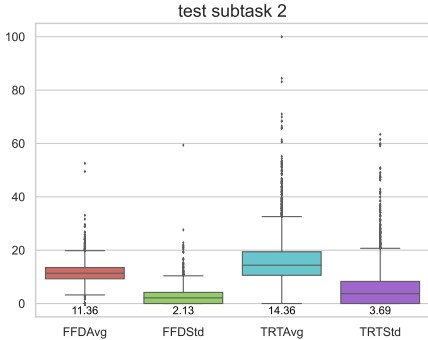

Figure 2: Boxplots showing the feature value distributions of the training data and the test sets of both subtasks. Below each box is the median value of each feature.

sentences read by 5 readers, extracted from the CopCo corpus (Hollenstein et al., 2022).

### 4.3 Preprocessing

**Tokenization** The tokens in the sentences are split in the same manner as they were presented to the participants during the reading experiments. Hence, this does not necessarily follow a linguistically correct tokenization. For example, the sequences "(except," and "don't" were presented as such to the reader and not split into "(", "except", "," and "do", "n't" as a tokenizer would do. It is the participants' decision how to deal with these tokens.

**Feature Extraction** The data contains scaled features in the range between 0 and 100 to facilitate evaluation via the mean absolute average (MAE). The eye-tracking feature values (FFDAVG and TR-TAVG) are averaged over all available readers of a corpus. This preprocessing step is done separately for each corpus before combining them. Table 2 shows the scaled features values across the full dataset of Subtask 1. In Figure 2, we present the distributions of the feature values for the training set and the test set of Subtasks 1 and 2. Finally, Figure 3 in the Appendix shows the individual plots for each language.

## 5 Evaluation

In this section, we describe the evaluation procedure used to assess the submitted predictions of the participating teams.

Any additional data source was allowed to train the models, as long as it is freely available to the research community. For example, additional eye-tracking corpora, additional features such as brain activity signals, pre-trained language models, etc.

### 5.1 Scoring Metric

The submitted predictions are evaluated against the real eye-tracking feature values using the mean absolute error (MAE) metric, a measure of errors between paired observations including comparisons of predicted ($y$) versus observed ($x$) values for each word in the test set:

$$MAE = \frac{\sum_{i=1}^{n} |y_i - x_i|}{n} \quad (1)$$

The winning system is defined as the one with the lowest average MAE across all 4 features. We reported additional metrics for analysis, namely $R^2$ for all features individually and aggregated, but only MAE was used for the ranking.

### 5.2 Mean Baseline

We use the mean central tendency as a baseline for this regression problem, i.e., we calculate the mean value for each feature from the training data and use it as a prediction for all words in the test data. Table 3 shows the MAE scores achieved by this mean baseline for each eye-tracking feature.

For Subtask 1, we add an additional stronger mean baseline calculated over the training set of each language individually. This baseline assumes that the language of each sentence is known to the

| Rank | Team Name | MAE | FFDAVG | FFDSTD | TRTAVG | TRTSTD | $R^2$ | Reference |
|------|-----------|-----|--------|--------|--------|--------|-------|-----------|
| 1 | HkAmsters | 3.01 | 4.40 | 4.15 | 1.76 | 1.73 | 0.61 | Salicchi et al. (2022) |
| 2 | DMG | 3.65 | 5.65 | 4.43 | 2.61 | 1.92 | 0.49 | Takmaz (2022) |
| - | Lang. baseline | 4.27 | 3.55 | 2.03 | 6.56 | 4.94 | 0.34 | - |
| 3 | NU HLT | 5.49 | 6.67 | 8.38 | 3.93 | 2.99 | -0.18 | Imperial (2022) |
| 4 | Poirot | 5.50 | 8.37 | 5.68 | 5.47 | 2.50 | -0.03 | Srivastava (2022) |
| 5 | UFAL | 5.72 | 8.81 | 5.73 | 5.77 | 2.58 | 0.00 | Bhattacharya et al. (2022) |
| - | Mean baseline | 5.73 | 8.82 | 5.89 | 5.69 | 2.54 | 0.00 | - |
| 6 | TorontoCL | 11.09 | 18.84 | 8.89 | 13.06 | 3.57 | -2.04 | |

Table 3: Overall results of **Subtask 1** showing the best submission per team and the mean baselines, including the overall MAE and $R^2$ scores, as well as the individual MAE scores for each feature. The teams are ranked by the MAE averaged across all five eye-tracking features (third column).

| Rank | Team Name | MAE | FFDAVG | FFDSTD | TRTAVG | TRTSTD | $R^2$ | Reference |
|------|-----------|-----|--------|--------|--------|--------|-------|-----------|
| 1 | Poirot | 4.23 | 5.60 | 5.65 | 2.95 | 2.73 | -0.26 | Srivastava (2022) |
| 2 | DMG | 4.97 | 6.90 | 5.77 | 5.45 | 1.73 | -0.57 | Takmaz (2022) |
| - | Mean baseline | 5.73 | 8.82 | 5.89 | 5.69 | 2.54 | 0.00 | - |
| 3 | NU HLT | 7.09 | 14.65 | 4.04 | 7.53 | 2.12 | -1.83 | Imperial (2022) |

Table 4: Overall results of **Subtask 2** showing the best submission per team and the mean baseline, including the overall MAE and $R^2$ scores, as well as the individual MAE scores for each feature. The teams are ranked by the MAE averaged across all five eye-tracking features (third column).

system. This second baseline was not reported in the rankings, but serves for further analysis.

# 6 Participating Teams & Systems

Six teams and a total of 37 participants registered on the competition website. All six teams submitted their predictions during the evaluation phase for Subtask 1. Four of the teams also submitted predictions for Subtask 2. Each team was allowed three submissions for each subtask during the evaluation phases. Finally, 5 teams published system description papers outlining their approaches (see Table 3 for all references).

## 6.1 Methods

The participating teams submitted predictions generated from various approaches, from regression algorithms such as random forests (NU HLT) and linear regression models (HkAmsters) with a wide range of lexical, cognitively and phonetically-motivated features (NU HLT), to neural approaches that fine-tune large pre-trained transformer models with additional regression heads, and integrate adapters into pre-trained transformer language models (DMG) (Pfeiffer et al., 2020; Han et al., 2021).

Some teams chose to build language-specific models (e.g., HkAmsters, DMG), while others merged the words from all languages into a common vocabulary space in which all words are converted to their IPA forms (NU HLT). Moreover, representations from both monolingual models such as GPT-2 (Radford et al., 2019) as well as multi-lingual transformer models such as mBERT (Devlin et al., 2019) and XLM (Lample and Conneau, 2019) were also included (Poirot). For the second subtask, dealing with a new unseen language was handled again through a common phonetic vocabulary space (NU HLT), through translation (i.e., translating the Danish text to German and then using a German model for prediction) (DMG), or zero-shot learning (Poirot).

# 7 Results

In this section, we describe the prediction performance achieved by the participating teams. The official results of this shared task are presented in Tables 3 and 4 for Subtask 1 and 2, respectively. The best results for the first subtask on multilingual prediction were achieved by Team HkAmsters with language-specific regression models based on word-level features such as word length, word frequency, and surprisal scores estimated with GPT-2 (Radford et al., 2019). For the second subtask on

cross-lingual prediction, the winning team (Poirot) used a zero-shot hybrid model based on a multilingual transformer embeddings as well as statistical features.

# 8  Outlook & Conclusion

We presented the results of the second shared task on predicting token-level eye-tracking features recorded during natural reading of sentences or longer text spans. In this second edition, we focused on multilingual and crosslingual prediction. We hope the CMCL Shared Task makes a lasting contribution to the field of linguistic cognitive modelling by providing researchers with a standard evaluation framework and a high quality dataset. Despite the limited size of the training and test sets as well as the diversity of text domains within the eye-tracking corpora, many previously reached conclusions can now be tested more thoroughly and future models can be compared on a shared multilingual benchmark.

## Acknowledgements

Emmanuele Chersoni acknowledges the Startup Fund for RAPs under the Strategic Hiring Scheme from the Hong Kong Polytechnic University (PolyU UGC, BD8S).

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

# A  Appendix

Figure 3 shows the distributions of the feature values for the data of all six languages.

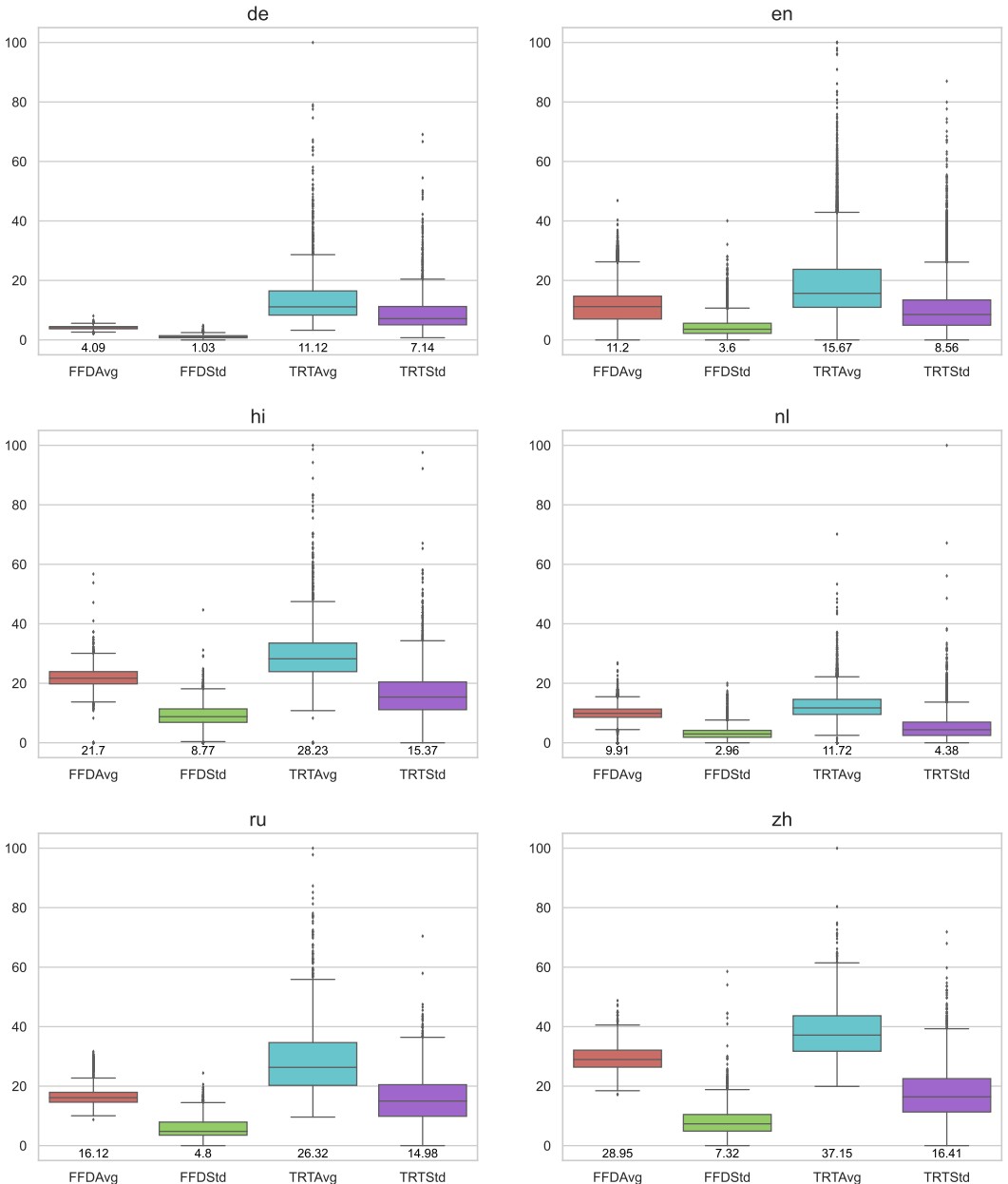

Figure 3: Boxplots showing the feature value distributions of the eye-tracking data of all languages of Subtask 1. Below each box is the median value of each feature.