# OpenReview forum: "CMCL 2022 Shared Task on Multilingual and Crosslingual Prediction of Human Reading Behavior"
_aclweb.org/ACL/2022/Workshop/CMCL_Shared_Task — CMCL Shared Task_

### Official Review · Reviewer_5fgw · 2022-03-20
**Good description of the multilingual eye-tracking feature prediction shared task at the CMCL workshop.**

**Rating:** 8
**Confidence:** 5

**Review:**

### Summary
This paper presents the second shared task on eye-tracking data prediction of the CMCL workshop. In this year's shared task, the participating teams submit systems to predict 4 eye-tracking features, including: the mean and the standard deviation of the first fixation duration (FFDavg and FFDstd), and the mean and standard deviation of the total reading time (TRTavg and TRTstd). The shared task contains two challenges. Subtask 1 requires the participants to predict the features in 6 provided langauges (zh, hi, ru, en, nl, de), and subtask 2 involves a new language (dk). This paper describes the dataset, preprocessing steps, the scoring metric, and the baseline methods. This paper also lists the results of the systems.

### Reasons to accept
- CMCL shared task is an important initiative. The datasets collected and the methods examined in CMCL shared task have a lot of potential future impacts.
- The version of this year introduces additional dataset, and considers how models trained on given languages can generalize to a held-out language.
- This paper describes the shared task, the data, and the participating systems clearly.

### Reasons to reject
I do not find reasons to reject this paper.

### Comments
- Abstract at second last line: "transformer" -> "Transformer"
- Capture of figure 1: "Example sentence" -> "An example sentence"
- The line above Secetion 7: "zer-o-shot learning" -> "zero-shot learning"
- The last bibliography entry at page 6 appears in blue font, while other entries appear in normal font. (Probably this is a problem with my display?)
- Lample & Conneau (2019) bib entry appears duplicated.
- Merkx and Frank (2021) bib entry appears duplicated.

---

### Official Review · Reviewer_LKSF · 2022-03-21
**The official shared task paper for CMCL 2022**

**Rating:** 9
**Confidence:** 4

**Review:**

The present presents the full description of the 2022 CMCL Shared Task which covers multilingual and crosslingual prediction of human reading behavior using diverse eye-tracking datasets. The paper illustrates the performance of each registered team with respect to their proposed (novel) solutions for both tasks evaluated via MSE. This paper merits a default acceptance as it summarizes the entire activity and contributions done for the Shared Task.

---

### Decision · Program_Chairs · 2022-03-28

Accept